# OpenReview forum: "Evaluating Copyright Takedown Methods for Language Models"
_NeurIPS.cc/2024/Datasets_and_Benchmarks_Track — NeurIPS 2024 Track Datasets and Benchmarks Poster_

### Official Review · Reviewer_VRnW · 2024-07-15

**Rating:** 7
**Confidence:** 3
**Correctness:** Yes
**Clarity:** Yes

**Review:**

The paper provides a comprehensive evaluation of the copyright takedown methods of LLM content generation, which is crucial for the development of the LLM models. However, the evaluation is only done on LLaMA 2, which may not be representative of all LLM models.  The evaluated methods are mostly about reducing the memorization of the copyrighted text, which may not be the only way to avoid copyright infringement.

**Strengths:**

Addressing the crucial issue of copyright in LLM content generation, the paper is timely and relevant. It contributes valuable insights to a significant area of concern in developing and deploying large language models.

The paper provides a comprehensive evaluation of the copyright takedown methods of LLM content generation, which contains the risk, utility, and efficiency of the model.

By exploring a range of takedown methods, including system prompts, decoding methods like MemFree, and unlearning methods like GA, the paper offers a broad perspective on available strategies. This diversity allows for a more nuanced evaluation of the strengths and weaknesses of each approach.

**Additional Feedback:**

N/A

**Documentation:**

Yes

**Ethics:**

The paper has declared the ethics concerns in the checklist.

**Limitations:**

A Limitations section is provided

**Opportunities For Improvement:**

Some of the previous works are not included, such as https://aclanthology.org/2023.emnlp-main.458/ and https://ojs.aaai.org/index.php/AAAI/article/view/29741

I suggest moving the experiment figures related to unlearning/RCAD to the main text. These results are important, and they deserve to be highlighted in the main body of the document. Currently, they are difficult to locate in the supplementary material.

The paper uses the BookSum dataset to evaluate copyright infringement. The BookSum dataset is collected from Project Gutenberg,  a collection of public domain books. As a result, the BookSum dataset is generally free for anyone to use, and there will be no copyright infringement issue of LLMs to memorize book contents in the BookSum dataset. This may not be a good indicator of the copyright infringement of the LLM models, as public domain books are more easily accessible and more likely to be used in the training data of the LLM models, resulting in a higher memorization rate that may differ from the copyrighted books.

The evaluation is only done on LLaMA 2, which may not be representative of all LLM models. The evaluation should include other mainstream LLM models such as GPT-4, Claude, etc. It may be a concern that most of the methods require the model parameters, which may not be available for all the models. However, using system prompts is still available for many models available only by the API.

The evaluated methods are mostly about reducing the memorization of the copyrighted text, which may not be the only way to avoid copyright infringement. For example, alignment methods can also be used to avoid copyright infringement, by letting the model refuse to generate the copyrighted text.

The evaluated prompt is limited to a hint, which may not be representative of all the prompt types.

**Relation To Prior Work:**

Some of the previous works are not included, such as https://aclanthology.org/2023.emnlp-main.458/ and https://ojs.aaai.org/index.php/AAAI/article/view/29741

**Summary And Contributions:**

The paper provides a comprehensive evaluation of copyright takedown methods in LLM content generation, which is crucial for the development of these models. It introduces three main goals for takedown methods: 1) avoiding the memorization of copyrighted text, 2) maintaining the quality of common text generation, and 3) ensuring low overhead to generation speed. The evaluation criteria include the risk of copyright infringement (measured by win rate), the utility of the model post-takedown (assessed using common evaluation datasets), and the efficiency of the model (evaluated by inference speed). The experiments offer insights into various takedown methods, including system prompts, decoding methods like MemFree, and unlearning methods like GA. Copyright concerns are a significant issue in LLM content generation, making the evaluation of these takedown methods essential for the models' development.

---

> ### Author Rebuttal · Authors · 2024-08-16
>
> Dear reviewer, thank you very much for your positive, insightful comments and for recognizing our comprehensive evaluation of copyright takedown methods. We would like to address the feedback and questions below, and we will edit the paper accordingly.
>
> ## **Response to Feedback 1 (Some previous works are not cited)**
> We thank the suggestion and will discuss the suggested previous work in the related work section.
>
> ## **Response to Feedback 2 (Move unlearning and R-CAD results to the main sections)**
> We appreciate the comment. We put the unlearning/RCAD results in the appendix due to the page limit. We will move them to the main section in our final version.
>
> ## **Response to Feedback 3 (BookSum may not be a good indicator of copyright infringement)**
> We thank the reviewer for their insightful comment regarding the use of the BookSum dataset in our evaluation. We agree that the memorization rates of public domain books may differ from what might be seen with copyrighted books. However, our primary focus is not to quantify how much copyrighted contents are memorized by current models, but rather to evaluate the effectiveness of mitigation methods when potential infringement happens. As a result, higher memorization rate is actually an advantage for such studies. Moreover, by employing fine-tuning and retrieval augmentation techniques, we deliberately induce the model to explicitly memorize and output the book contents. This allows us to simulate the worst-case scenarios of potential copyright infringement, regardless of the original memorization rates in the pre-trained model. BookSum dataset provides a useful proxy for evaluating copyright takedown methods to navigate the complex legal landscape surrounding copyrighted material.
> - Benchmark consistency: Using public domain works allows us to release our benchmark datasets, promoting reproducibility and further research in this area.
> - Legal and ethical considerations: Utilizing copyrighted books makes it easier to share the data with other researchers, promoting open research on this important problem.
> - Structural similarity: Despite being in the public domain, these books likely share structural and linguistic similarities with copyrighted works, allowing us to examine memorization patterns in long-form textual content.
>
> ## **Response to Feedback 4 (More models should be tested instead of only Llama2)**
> We acknowledge that we need to evaluate more models. We also did experiments on DBRX-instruct (See Appendix E.3) and observed similar results. We add a new set of experiments on Gemma-2-9B-it model (see our common rebuttal for all reviewers). The new results show our conclusions hold across different LMs.
>
> ## **Response to Feedback 5 (Methos evaluated are not enough)**
> We have included one alignment strategy in our experiments, in which we use DPO to train the model to refuse to answer the question that contains the blocklisted content (See $\text{Unlearning}_{\text{PO}}$ in Appendix C.2 lines 659-662), and we find that similar to other unlearning methods, using DPO also sacrifices a lot utility when it becomes effective. It also requires extensive hyperparameter search for better performance. Additionally, system prompts already test some amount of general refusal capability and we use already aligned chat models. We agree that alignment is an interesting area of further exploration and future work can expand evaluation to more complex alignment strategies.
>
> ## **Response to Feedback 6 (Eval prompt may not be representative)**
> We acknowledge that the prompt for evaluation cannot represent all the cases when there’s a copyright issue. The goal of using hint is to simulate a “worst case” scenario, in which the model is heavily induced to generate the copyrighted content, we believe if a takedown method is good at handling the scenario used in our pipeline, then it will also perform well in other scenarios, in which the risk of copyright issues is likely less than ours. In fact, these sorts of prompts are well-represented in ongoing litigation (e.g., New York Times v. OpenAI) in demonstrating infringement. With this prompt, we observe a significant rate of regurgitation when no mitigations are applied, which means our setting provides a good starting point for evaluating the takedown methods. An interesting area of future research would be to collect “in the wild” extraction prompts from real users, but this is beyond the scope of this paper. We will, however, add this to the discussion in the paper.

---

> > ### Comment · Reviewer_VRnW · 2024-08-18
> >
> > Thanks the authors for their response. I have read the response and the other reviews. Most of my concerns have been addressed in the response.
> >  I have updated my assessment of the paper.

---

### Official Review · Reviewer_UkhZ · 2024-07-25

**Rating:** 6
**Confidence:** 4
**Correctness:** Yes
**Clarity:** Yes

**Review:**

The paper provide a comprehensive evaluation, despite some minor issues.

**Strengths:**

1. The problem studies in this paper is important in real-world application.

2. The writing and introduction for takedown methods is comprehensive.

3. Evaluation includes multiple metrics for infringement risk, utility preservation, and computational efficiency.

**Additional Feedback:**

N/A

**Documentation:**

Yes

**Limitations:**

Please see weakness

**Opportunities For Improvement:**

1. More model should be tested instead of only LLAMA2.

2. Unlearning time cost should be evaluated.

**Relation To Prior Work:**

Yes

**Summary And Contributions:**

This paper proposes a framework for evaluating methods to prevent language models from generating copyrighted content. The evaluation focuses on three main criteria: reducing infringement risk, maintaining utility, and ensuring efficiency.

---

> ### Author Rebuttal · Authors · 2024-08-16
>
> We thank the reviewer's positive, insightful comments and for recognizing our thorough evaluation of copyright takedown methods. We address the feedback and questions below, and we will update the paper accordingly.
>
> ## **Response to Feedback 1 (More models should be tested instead of only Llama2)**
>  We also did experiments on DBRX-instruct (See Appendix E.3) and observed similar results. We add a new set of experiments on Gemma-2-9B-it model (see our common rebuttal for all reviewers). The new results show our conclusions hold across different LMs, and we will include these results in the final version of the paper.
>
> ## **Response to Feedback 2 (Unlearning time cost should be evaluated)**
> We thank the reviewer for the comment. As we noted in the submission (See Section 3.2 lines 197-199), the cost of each intervention consists of both online (the cost at inference time) and offline cost (the cost of training). We will make this clearer in the paper.
>
> | Cost Type                                                    | Analysis                                                     |
> | :----------------------------------------------------------- | :----------------------------------------------------------- |
> | Online Cost (The cost incurred during each model interaction. e.g., modification to the decoding process) | Zero. Current unlearning methods do not modify the inference stage of models. |
> | Offline Cost (One time cost like modifying the model or database) | ~ 30 GPU hours for each method to search the best hyperparameter combination across 25 candidates (See Appendix B.1 in lines 700-708), indicating that getting an ideal hyperparameter for unlearning methods requires an extremely high offline cost. |

---

### Official Review · Reviewer_ZFB7 · 2024-07-25
**Evaluation framework for copyright takedowns**

**Rating:** 8
**Confidence:** 3

**Review:**

The paper's strengths lie in addressing a novel and critical issue in AI and copyright law with the introduction of the CoTaEval framework, which provides a comprehensive and robust evaluation of various takedown methods. The study's experimental setup is good for reproducibility and offers insights into the trade-offs between infringement risk reduction, utility preservation, and efficiency. However, the evaluation is limited by a narrow dataset scope and incomplete mitigation of infringement risks. For more details, see below.

**Strengths:**

- The paper tackles a pressing issue in the field of AI, law and ethics, providing a structured approach to evaluate copyright takedown methods in language models
- There is a robust and well-thought-out framework that considers multiple dimensions of evaluation, including infringement risk, utility, and efficiency
- The framework examines a range of methods (system prompts, decoding-time interventions, unlearning approaches), offering a broad perspective on potential solutions
- The authors provide a thorough description of their experimental setup, including the datasets used (NewsQA and BookSum), evaluation metrics, and the models tested
- The results clearly show the trade-offs involved in different takedown methods, providing valuable insights for future research and development in this area

**Additional Feedback:**

A minor comment to the authors. Section 3 is titled “The CoTaEval Evaluation pipeline”. Since CoTaEval includes “Evaluation”, the second use of “Evaluation” is redundant. But breaking these style rules might be warranted here if the authors want to improve readability,

**Clarity:**

The paper is well-written and clearly structured. Each section logically follows from the previous one, making it easy to follow as a reader.

**Correctness:**

The evaluation methods and experimental design are appropriate. The detailed description of the experimental setup and metrics should ensure reproducibility and reliability.

**Documentation:**

The supplementary materials and appendices provide additional details on the datasets, metrics, and experimental setup. The online repository with code and further details makes it even more accessible.

**Ethics:**

The paper discusses the ethical implications of copyright infringement by language models and the necessity of takedown methods. However, the exploration of the potential negative impacts of these methods on the broader AI community and the users of these models could be expanded, but would have to come at the cost of reducing other sections. It is not a critical requirement in my view.

**Limitations:**

The authors acknowledge several limitations, including the relatively small evaluation datasets and the need for more diverse general utility evaluations. However, they could also discuss the offline cost of implementing some takedown methods, which could be a significant barrier to their adoption. Additionally, the discussion on potential negative societal impacts is brief and could benefit from a more in-depth analysis, especially regarding how these methods might affect the broader deployment of language models.

**Opportunities For Improvement:**

- The evaluation focuses on news articles and books, which may not fully represent the variety of copyrighted content language models are exposed to. Including more diverse datasets could provide a more comprehensive assessment. However, it is a good place to start.
- The unlearning methods are noted to be highly sensitive to hyperparameters, requiring extensive tuning. This challenge could be better addressed or mitigated in the paper.
- Although not a requirement for this work, the paper could explore hybrid methods or more sophisticated techniques
- The paper could go deeper into discussions on the scalability of these methods for larger models or real-world applications, which is critical for practical deployment.

**Relation To Prior Work:**

The paper effectively situates itself within the existing literature, referencing relevant prior work. However, it could benefit from a deeper comparison with similar evaluation frameworks in other, but related, domains.

**Summary And Contributions:**

The authors present an evaluation framework designed to assess the effectiveness of various copyright takedown methods in language models. The methods examined include system prompts, decoding-time filtering interventions, and unlearning approaches. The framework evaluates these methods based on their ability to prevent the generation of copyrighted content, retain uncopyrightable factual knowledge, and maintain the model’s utility and efficiency.

Main contributions:

- Introduction of the COTAEVAL evaluation framework.
- Taxonomy of copyright infringement causes and takedown methods.
- Comprehensive evaluation of takedown methods using NewsQA and BookSum datasets.
- Analysis of the trade-offs between infringement risk reduction, utility preservation, and computational efficiency.

---

> ### Author Rebuttal · Authors · 2024-08-16
>
> We thank the reviewer's positive, insightful comments and for recognizing our thorough evaluation of copyright takedown methods. We address the feedback and questions below, and will update the paper accordingly.
> ## **Response to Feedback 1 (News and Books cannot fully represent the variety of copyrighted content)**
> We acknowledge that news and books cannot fully represent the variety of copyrighted content.  Our choice of these specific types of content is guided by several factors:
> - Recent litigation (like New York Times vs Microsoft) highlights the relevance of these mediums.
> - Their higher availability, especially on a larger scale.
> - The feasibility of evaluating the model’s blocklisted and in-domain utility. For news, we ask the model to answer the context-related question, while for books, we ask the model to summarize the context. (See line 223) on these datasets.
>
> We believe that starting our evaluations with news and books provides a solid foundation for evaluating copyright takedown methods. We recognize the importance of including a variety of copyrighted content and have discussed this limitation in Appendix A.1 (See lines 556-557).  Future work can build on our foundation for a more diverse collection of copyrighted content such as code and lyrics.
>
> ## **Response to Feedback 2 (The unlearning methods need to be better addressed)**
> We acknowledge the limitation that the unlearning methods evaluated in our pipeline are highly sensitive to hyperparameters, and have discussed it in the submission (lines 294-297). To alleviate this limitation, we conducted a comprehensive hyperparameter search (25 combinations for each method), which translates to 30 GPU hours (see lines 700-708). We will make this discussion clearer in the paper.
>
> ## **Response to Feedback 3 (Exploration for hybrid takedown methods)**
> We appreciate this suggestion and agree that this is a very important direction for future studies.
>
> ## **Response to Feedback 4 (Scalability discussion)**
> This is a great point. We also provide a short discussion below for the main challenges for scaling up methods in our evaluation:
>
> | Category                | Method             | The main challenges in scaling up                            |
> | ----------------------- | ------------------ | ------------------------------------------------------------ |
> | Generic Prevention      | System Prompt      | The deployment cost of adding system prompt is not dependent on the size of blocklisted content and the model size. However, results for the current scale have suggested the method cannot completely prevent undesired regurgitation, and when the dataset scales up, the number of failure cases may increase. |
> | Generic Prevention      | Top-k Perturbation | The deployment cost and performance of Top-k perturbation are not dependent on the size of blocklisted content and the model size. However, results for the current scale have suggested the method has a poor performance. |
> | Decoding time Takedown  | MemFree            | When the blocklisted content scales up, it will require a larger bloom filter, which incurs a higher building, storing, and lookup cost, and may lead to efficiency issues. Though there may be efficiency improvements around the implementation of the bloom filter. |
> | Decoding time Takedown  | R-CAD              | R-CAD has a lower efficiency cost than MemFree since it does not require a per-token lookup penalty. However, there is still some cost as the size of the blocklist is increased. This can be mitigated with an efficient retriever and index. |
> | Training-based Takedown | Unlearning         | The performance of unlearning methods will likely be worse when scaled up. Existing work [1] has shown that as the forget set scales up, it becomes harder for the model to maintain its utility. |
>
> [1] Maini, Pratyush, et al. "Tofu: A task of fictitious unlearning for llms." *arXiv preprint arXiv:2401.06121* (2024).

---

### Author Rebuttal · Authors · 2024-08-16

We appreciate the reviewers' strong support for the contributions of the paper and their insightful comments.  Based on the suggestions by Reviewers UkHz & VRnW, we conducted additional evaluation and assessed the copyright takedown methods on the Gemma-2-9B-it [1] model in the RAG setting and reported the results in the attached PDF file.

Similar to the results of the Llama2-7B-chat and Llama2-70B-chat model, we observe that:
1. Adding system prompt and MemFree cannot completely prevent the model from generating the blocklisted content, and the effectiveness of the system prompt is worse than Llama-2 model.
One possible reason is that Gemma-2-9B-it model is not trained with the system prompt and does not have specific tokens to denote the start and end of the system prompt. In our evaluation, we directly concatenate it with the user prompt as the new input to the model.
2. Top-k perturbation will greatly hurt the model’s utility when it becomes effective.
We will incorporate the results in the next version of the paper.​

[1] Team, Gemma, et al. "Gemma 2: Improving open language models at a practical size." arXiv preprint arXiv:2408.00118 (2024).

---

### Decision · Program_Chairs · 2024-09-26

**Decision:**

Accept (Poster)

**Comment:**

- Summary (borrowed from reviewer VRnW)
    - The paper provides a comprehensive evaluation framework, named as CoTaEval, to evaluate copyright takedown methods in LLM content generation, which is crucial for the development of these models. It introduces three main goals for takedown methods: 1) avoiding the memorization of copyrighted text, 2) maintaining the quality of common text generation, and 3) ensuring low overhead to generation speed. The evaluation criteria include the risk of copyright infringement (measured by win rate), the utility of the model post-takedown (assessed using common evaluation datasets), and the efficiency of the model (evaluated by inference speed). The experiments offer insights into various takedown methods, including system prompts, decoding methods like MemFree, and unlearning methods like GA. Copyright concerns are a significant issue in LLM content generation, making the evaluation of these takedown methods essential for the models' development.
- Overall
    - All reviewers agreed on that this paper addresses important and timely problem for copyright infringement risks. This evaluation framework is regarded as valuable in the sense that it contains evaluation dataset, criteria, and metrics. Under the framework three takedown methods are experimentally evaluated, which would be baselines of future research.
    - One of the major common concerns was that evaluations were conducted only on Llama-2 models. However, the authors complemented additional experimental results on Gemma-2-9B-it model, showing similar trends with Llama-2 models. Other concerns have been addressed during the discussion phase, such as limited scope of the evaluation datasets, evaluated methods, and discussions on unlearning methods. These points should be included in the final version of the manuscript.